# Plasmapheresis in the ICU

**DOI:** 10.3390/medicina59122152

**Published:** 2023-12-12

**Authors:** Guleid Hussein, Bolun Liu, Sumeet K. Yadav, Mohamed Warsame, Ramsha Jamil, Salim R. Surani, Syed A. Khan

**Affiliations:** 1Mayo Clinic Health System, Mankato, MN 56001, USA; liu.bolun@mayo.edu (B.L.); yadav.sumeet@mayo.edu (S.K.Y.); warsame.mohamed@mayo.edu (M.W.);; 2Sindh Medical College, Jinnah Sindh Medical University, Karachi 75510, Pakistan; ramsha.jamil@ymail.com; 3Department of Anesthesiology, Mayo Clinic, Rochester, MN 55905, USA; 4Department of Medicine and Pharmacology, Texas A&M University, College Station, TX 77843, USA

**Keywords:** plasmapheresis, ICU, TPE

## Abstract

Therapeutic plasma exchange (TPE) is a treatment paradigm used to remove harmful molecules from the body. In short, it is a technique that employs a process that functions partially outside the body and involves the replacement of the patient’s plasma. It has been used in the ICU for a number of different disease states, for some as a first-line treatment modality and for others as a type of salvage therapy. This paper provides a brief review of the principles, current applications, and potential future directions of TPE in critical care settings.

## 1. Introduction

Therapeutic plasma exchange (TPE), also known as plasmapheresis, is an extracorporeal technique that replaces patients’ plasma to remove pathogenic molecules. The common targets for removal are autoimmune antibodies, donor-specific antibodies, excessive paraproteins, cytokines, and endogenous and exogenous toxins [1]. TPE has become a commonly used, life-saving standard therapy for various conditions in intensive care settings. In this paper, we will introduce the mechanisms and principles of TPE, briefly describe some of the common disease states prompting TPE in the ICU, discuss some of its inherent challenges in the ICU, and provide potential future directions for the treatment paradigm.

## 2. Mechanisms and Principles of Therapeutic Plasma Exchange (TPE)

During a TPE session, the patient’s blood is drawn a peripheral or central access site into the apheresis system. TPE is performed via two different major techniques: centrifugal separation and membrane separation [2]. In the centrifugal separation method, the blood is spun in an apheresis device and the different components are separated via specific gravity. In the membrane separation method, the blood crosses non-selective microporous membranes, which allow for the passage of molecules less than a certain weight and also allows for the retention of blood cells. The respective levels of efficacy of these two methods are comparable based on historical studies, although the centrifugal process was reported to be more time-efficient [3]. Notably, the membrane separation method is more commonly used in Japan and Germany, while the centrifugal separation method is dominant in the USA and the rest of Europe [4]. After the apheresis process, the blood is infused back into the patient along with healthy donor plasma or albumin (with or without normal saline).

Most TPE methods use albumin replacement to ensure less immunogenicity and improved safety. Certain hematological emergencies require replacement with plasma and cryoprecipitate to restore coagulation factors and normal coagulation function, such as thrombotic thrombocytopenic purpura (TTP), thrombotic microangiopathy (TMA) with factor H autoantibody, drug-induced TMA, and ANCA vasculitis-associated diffuse alveolar hemorrhage (DAH) [5]. 

TPE nonspecifically removes plasma molecules, including pathogenic molecules, normal coagulation factors, and immunoglobulin, which could put patients at higher risk of thrombosis, bleeding, infection, or allergic reaction (exclusively from plasma). Based on a single-center retrospective study in France, less than half of TPE sessions had at least one adverse effect, with hypocalcemia (19.6%) and hypotension (15.2%) being the most common; severe adverse effects only happened in 5.4% of patients [6]. Immunoadsorption (IA) was developed in the 1990s by using a specific plasma filter with bound antigens to target the immunoglobulin and preserve other plasma components. This approach could potentially minimize the risk of complication, and it is under investigation as a substitute for TPE in certain conditions [7,8].

ICU patients undergoing TPE can have multiple organ failure and require other extracorporeal supporting systems, such as intermittent hemodialysis (IHD), continuous renal replacement therapy (CRRT), and extracorporeal membrane oxygenation (ECMO). Such procedures could be performed sequentially or simultaneously via single or multiple access points. Systems could be combined in series, parallel, or hybrid mode [9]. 

## 3. Indications for Therapeutic Plasma Exchange in the ICU

Since 1986, the American Society for Apheresis (ASFA) has comprehensively reviewed the scientific evidence of the indications of therapeutic apheresis and issued detailed guidelines. The most recent ninth edition was published in early 2023 and highlighted 77 diseases with 119 indications for therapeutic plasma exchange [5]. There are 20 indications with TPE listed as a first-line treatment (ASFA category I) and 23 indications with TPE listed as a second-line treatment (ASFA category II) for various diseases (see Table 1). The ‘Kidney Disease: Improving Global Outcomes’ (KDIGO) group and American Academy of Neurology (AAN) also provide guidelines covering a few specific indications under their specialties [10,11]. The British Society for Hematology and the Japanese Society for Apheresis also issued guidelines reflecting the local experts’ consensus [12,13]. A great number of patients who require TPE are critically ill and will require intensive care monitoring. Guidelines have recommended individualized approaches based on the patient’s condition and following the local organizational policy based on resources and standard practice.

The optimal timing of initiation of TPE varies significantly by condition, from emergent use (within 4–6 h) to urgent use (within 24 h) to planned routine treatment. Intensivists should carefully weigh the risks and benefits of TPE based on the clinical context. TPE should be initiated as early as possible in combination with other treatment modalities for organ- or life-threatening conditions, such as TTP, catastrophic antiphospholipid syndrome (CAPS), mushroom intoxication, symptomatic hyperviscosity syndrome, severe myasthenia gravis, fulminant Wilson’s disease, and diffuse alveolar hemorrhage caused by autoimmune disease [14,15,16,17]. For devastating neurological conditions, such as Guillan-Barre/acute inflammatory demyelinating polyneuropathy and N-methyl-D-aspartate receptor antibody encephalitis, early initiation to stop ongoing injury processes could prevent permanent damage to the neurological system and may lead to better outcomes [18]. 

Since the COVID-19 pandemic, TPE has also been investigated for severe COVID-19 infection in the hope of removing excessive cytokines to alleviate significant inflammation and cytokine release syndrome. There are retrospective studies and a pilot randomized controlled trial showing potential survival benefits in severely ill patients [19,20,21]. However, this topic remains controversial. The continued fluctuation of epidemics calls for more scientific evidence regarding TPE use in critical COVID-19 patients.

## 4. Plasmapheresis in the ICU 

### 4.1. Neurological Disorders

#### 4.1.1. Guillain–Barré Syndrome (GBS)

Guillain–Barré syndrome, or acute demyelinating polyneuropathy (AIDP), is a rare but severe autoimmune disorder characterized by the immune system mistakenly attacking the peripheral nervous system. Plasmapheresis is a critical intervention for GBS in the ICU. It effectively removes circulating autoantibodies and inflammatory molecules responsible for nerve damage, reducing the immune response and inflammation. Plasmapheresis is primarily used when patients with GBS experience rapidly progressive symptoms or severe forms of the disease, especially when the respiratory muscles are affected. It serves as a life-saving intervention in critical GBS cases.

There have been large trials demonstrating the efficacy of plasmapheresis in improving muscle strength, decreasing the necessity for mechanical ventilation, and quickening the recovery time for patients with severe GBS. For example, one of the earliest trials, the Guillain–Barre Syndrome Study Group, found that when plasmapheresis was compared to conventional therapy in 245 patients, there were statistically significant differences in improvement at four weeks, time to improve one clinical grade, time-independent walking, and outcome at 6 months [22]. In a 2017 meta-analysis that included six randomized, controlled trials and 649 individuals diagnosed with Guillain–Barré syndrome (GBS), it was found that the utilization of plasma exchange (PLEX) as a treatment was more effective when compared to providing supportive care alone [23]. Among the 623 GBS patients analyzed, those who underwent PLEX treatment had a lower likelihood of needing mechanical ventilation (14 percent as opposed to 27 percent), and they were also more likely to experience functional improvement within four weeks (57 percent compared to 35 percent) when compared to those who received only supportive care [23]. 

#### 4.1.2. Myasthenia Gravis (MG)

Myasthenia gravis is a chronic autoimmune disorder affecting neuromuscular junctions, leading to muscle weakness and fatigue. Plasmapheresis is a valuable strategy in the ICU. It removes circulating autoantibodies that block acetylcholine receptors at neuromuscular synapses, improving nerve-to-muscle signaling. Plasmapheresis is employed during MG exacerbations, particularly when patients experience severe muscle weakness or respiratory muscle involvement. It leads to rapid symptom relief, making it crucial in managing acute MG crises and preventing respiratory failure.

A study on the safety of plasma exchange in the treatment of MG found that plasma exchange was safe, effective, and well-tolerated [24].

#### 4.1.3. Chronic Inflammatory Demyelinating Polyneuropathy (CIDP)

CIDP is a chronic autoimmune disorder affecting the peripheral nerves, characterized by demyelination and inflammation. In the ICU, plasmapheresis is used strategically. It reduces autoimmune responses and inflammation by eliminating autoantibodies and inflammatory molecules from the plasma. Plasmapheresis is considered for CIDP patients with rapid deterioration, severe muscle weakness, or exacerbations to manage inflammation and reduce disability. It promptly improves the clinical outcomes and reduces disability in CIDP patients.

Two small randomized clinical trials have found that plasma exchange is effective for the treatment of CIDP [25,26]. In a meta-analysis summarizing these clinical trials, approximately two-thirds of patients treated with plasma exchange showed meaningful short-term clinical improvement as measured by their Neuropathy Impairment Score (see Appendix A) [27]. 

In conclusion, plasmapheresis plays a crucial role in the ICU’s management of immune-mediated neurological disorders such as GBS, myasthenia gravis, and chronic inflammatory demyelinating polyneuropathy (CIDP). Its use should be judiciously considered on a case-by-case basis in consultation with healthcare specialists to ensure optimal patient care.

#### 4.1.4. Lambert–Eaton Myasthenic Syndrome (LEMS)

LEMS is a rare autoimmune disorder characterized by muscle weakness and coordination difficulties resulting from an immune attack on calcium channels at neuromuscular junctions.

In the ICU setting, where patients with LEMS may present with severe muscle weakness that can significantly impact their daily activities and respiratory function, plasmapheresis can offer critical relief. Its mechanism of action involves the extraction of autoantibodies targeting voltage-gated calcium channels. By reducing the levels of these autoantibodies, plasmapheresis is used to improve neuromuscular signaling, thereby enhancing muscle strength and function.

While not a first-line treatment, plasmapheresis is considered when rapid improvement is paramount. This is particularly relevant in the ICU, where patients with severe LEMS may require immediate intervention. Plasmapheresis is usually administered as an adjunctive therapy alongside other treatments such as immunosuppressants (such as corticosteroids) or immunomodulatory medications. For example, in a trial by Dau et al., plasmapheresis combined with immunosuppressive therapy was more effective than either plasmapheresis or immunosuppressive therapy alone [28]. This combined approach aims to address the underlying autoimmune component of LEMS comprehensively.

It is important to note that the effects of plasmapheresis are temporary, necessitating periodic repetitions to sustain benefits. Regular monitoring by a multidisciplinary healthcare team is crucial when employing plasmapheresis in LEMS. This ensures its effectiveness and allows for timely adjustments to the treatment plan. Additionally, close observation is essential to manage any potential risks or complications associated with the procedure.

#### 4.1.5. Multiple Sclerosis (MS)

Plasmapheresis is typically reserved for acute exacerbations of MS that do not respond adequately to high-dose corticosteroid treatment. It may be considered in cases where rapid improvement is critical, such as when patients experience severe relapses with significant disability. 

In the sole officially documented clinical trial, 22 individuals suffering from CNS demyelinating disease (including 12 with MS) were randomly allocated to receive either plasma exchange or a placebo treatment. During the study, a moderate or significant enhancement in neurological disability was observed in 8 out of 19 (42%) instances of actual treatment, in contrast to just 1 out of 17 (6%) instances of the placebo treatment [29]. The main analysis yielded favorable results. Improvements were noted early in the treatment period and persisted during follow-up. Nevertheless, 4 patients who initially responded to the active treatment faced new episodes of demyelinating disease over a 6-month follow-up period. Only 2 out of 13 patients who did not show improvement during the treatment phase experienced a moderate or significant improvement during follow-up. Plasma exchange contributes to significant neurological recovery in a substantial number of patients severely impacted by acute episodes of idiopathic inflammatory demyelinating disease.

Based mainly upon the results of this trial, the guidelines from the American Academy of Neurology state that plasmapheresis should be considered for the adjunctive treatment of exacerbations in patients with relapsing forms of MS.

In an earlier trial by Weiner et al., 116 patients in a multicenter, randomized, double-blind controlled trial were enrolled in an 8-week course of 11 plasma exchange treatments in exacerbations of MS. The control group received sham therapy, and both groups received identical treatments with intramuscular ACTH and oral cyclophosphamide. The serum IgG levels decreased in the plasma exchange and sham treatment groups by 76% vs22% by treatment 5 and by 64% vs. 14% by treatment 11 [30]. 

#### 4.1.6. Neuromyelitis Optica Spectrum Disorder (NMOSD)

Neuromyelitis optica spectrum disorder (NMOSD) is a rare autoimmune demyelinating disease of the central nervous system, primarily affecting the optic nerves and spinal cord. It is characterized by recurrent and severe attacks of inflammation that lead to optic neuritis, causing vision impairment and transverse myelitis, resulting in paralysis and sensory disturbances. NMOSD is distinct from multiple sclerosis due to the specific involvement of the optic nerves and spinal cord and is associated with the presence of anti-aquaporin-4 antibodies in the majority of cases. Early and accurate diagnosis is crucial for appropriate management and to prevent further neurological disability. The treatment typically involves immunosuppressive therapies to control inflammation and prevent relapses.

In patients with severe symptoms or vision loss refractory to glucocorticoids, therapeutic plasmapheresis is suggested.

Non-randomized studies suggest that patients with severe symptoms benefit more when plasmapheresis, used alongside glucocorticoids, is initiated early. In a retrospective study by Bonnan et al., 60 patients were included, covering 115 attacks. These patients received plasma exchange (PLEX) typically within 7 days (ranging from 0–54 days) of the onset of clinical symptoms. The main goal was achieving complete improvement, while the secondary outcomes were categorized as either poor or good based on the upper or lower third of changes in the Expanded Disability Status Scale (EDSS) (comparing late-stage to baseline EDSS) [31]. The likelihood of achieving complete improvement steadily decreased from 50% with plasma exchange administered on day 0 to only 1–5% after day 20. The study’s multivariate analysis indicated that both the severity of initial impairment and the delay in receiving plasma exchange were significant factors influencing the chance of complete improvement (odds ratio of 5.3; 95% confidence interval range of 1.8 to 15.9). A lower baseline impairment level seemed to be correlated with the primary outcome. Additionally, reducing the delay in administering PLEX was found to positively affect the good secondary outcome but not the poor secondary outcome [31].

### 4.2. Hematological and Autoimmune Conditions

#### 4.2.1. Catastrophic Antiphospholipid Syndrome (CAPS) 

Catastrophic antiphospholipid syndrome (CAPS) is a form of antiphospholipid syndrome or APS characterized by extensive and often rapid microthrombus formation affecting multiple organ systems spanning the kidneys, lungs, brain, and heart, causing complications such as renal insufficiency, pulmonary embolism, and encephalopathy. Factors such as infections and surgery can instigate it and affect those who have antiphospholipid antibodies [15].

The exact mechanism behind the condition remains elusive; however, it is surmised that a genetic predisposition, environmental triggers, and antiphospholipid antibodies contribute to its etiology and pathogenesis. The treatment and management comprise a triple therapy approach, including anticoagulation, corticosteroids, therapeutic plasmapheresis (TPE), and immunoglobulins. The CAPS registry cites research that this combined approach is associated with higher survival rates [15,32].

TPE, an essential component of CAPS treatment, is used for the removal of harmful antibodies, complements, and cytokines from the blood. Multiple TPE sessions are required with concurrent laboratory monitoring. The TPE process usually involves treating 1 to 1.5 times the patient’s plasma volume (TPV) daily or every other day, using replacement fluid consisting of plasma alone or in combination with albumin. The TPE protocols may be refined in the future to involve and alternative treatments may be explored for refractory CAPS, including rituximab, cyclophosphamide, and eculizumab [5,15,32,33].

#### 4.2.2. Thrombotic Thrombocytopenic Purpura (TTP)

Thrombotic thrombocytopenic purpura is a life-threatening blood disorder caused by a marked deficiency in ADAMTS13 activity resulting in thrombocytopenia, microangiopathic hemolytic anemia, and multi-organ dysfunction. An immediate diagnosis and prompt treatment, often prior to confirmation of the ADAMTS13 deficiency, are the cornerstones of its management. 

Immune-mediated TTP (iTTP), a subset of TTP, is characterized by autoantibodies to ADAMTS13. These antibodies are often IgG and result in a disruption of the ADAMTS13 activity, resulting in aberrantly large von Willebrand factor multimers and consequent abnormal clot formation [34].

As mentioned previously, a prompt diagnosis and treatment are important in its management. Since all medical facilities are not able to perform in-house ADAMTS13 testing, diagnostic tools, including the PLASMIC and French scores, are used to assess the probability of TTP pending ADAMTS13 test results. High-risk PLASMIC score patients benefit significantly from TPE, whereas in low-to-intermediate-risk patients, the difference in survival outcomes between those undergoing TPE and those who do not is minimal [35].

The rapid initiation of therapeutic plasma exchange (TPE) is of paramount importance. TPE allows for the removal of autoantibodies while maintaining the ADAMTS13 activity. Corticosteroids are frequently used as an adjunct to TPE. Rituximab, an immunosuppressive agent, has also been used, as it has shown potential in preventing relapses. Caplacizumab has been administered alongside TPE and immunosuppression and is thought to aid in speeding the resolution of TTP and decreasing the risk of relapse [34,35]. 

Refractory cases pose a challenge; however, the alternative therapies include bortezomib, cyclosporine A, and cyclophosphamide. In addition, the current research is exploring recombinant ADAMTS13 and inhibitors of VWF–glycoprotein Ib/IX interaction as possible treatments [34,35].

#### 4.2.3. ANCA-Associated Vasculitis (AAV)

ANCA-associated vasculitis (AAV) consists of a set of rare autoimmune diseases that cause inflammation and consequent damage to small blood vessels, primarily in the kidneys and the lungs. They include microscopic polyangiitis (MPA), granulomatosis with polyangiitis (GPA), and eosinophilic granulomatosis with polyangiitis (EGPA). Glucocorticoids and cyclophosphamide are the standard therapy for inducing remission in ANCA-associated vasculitis (AAV). Rituximab can be used as an alternative treatment agent. TPE is mainly used for severe renal involvement or high-risk scenarios [36]. Earlier studies, such as the MEPEX trial [37], supported the efficacy of plasmapheresis, especially in cases of severe rapidly progressive glomerulonephritis in AAV; however, the result of the PEXIVAS trial showed no significant difference in mortality or renal death (end-stage kidney disease) between the TPE and non-TPE groups [37]. Despite the conflicting results, plasmapheresis continues to be considered a salvage treatment modality for patients with severe AAV who show little to no response to standard therapy. In addition, the KDIGO guidelines recognize TPE as a viable treatment option for severe cases of AAV with conditions such as diffuse alveolar hemorrhage [37].

#### 4.2.4. Hyperviscosity Syndrome (HVS)

Hyperviscosity syndrome (HVS) is characterized by a classic triad of clinical symptoms: neurological deficits, visual changes, and mucosal bleeding [38]. This condition is typically associated with an abnormal increase in serum proteins, which can be either monoclonal or polyclonal, or an increase in whole-blood components such as red blood cells (RBCs) and white blood cells (WBCs). Mucosal bleeding results from impaired platelet function. The common manifestations include epistaxis (nosebleeds), bleeding gums, and gastrointestinal bleeding. HVS can lead to a range of neurological symptoms, including headaches, stupor, dizziness, ataxia (loss of coordination), and seizures. These symptoms occur due to reduced blood flow to the brain resulting from increased blood viscosity. The visual symptoms may manifest as blurry vision, retinopathy (abnormalities in the retina), intraocular hemorrhages, and exudates. Ophthalmologic examinations play a critical role in the diagnosis, helping identify distinctive signs such as sausage-shaped retinal veins and retinal vein engorgement. The diagnosis of HVS relies primarily on a clinical evaluation, incorporating thorough history-taking and physical examination findings. Blood tests may reveal elevated levels of serum proteins or other blood components, which can support the diagnosis. An ophthalmologic assessment is a crucial component of the diagnostic process, enabling the identification of retinal abnormalities associated with HVS [38].

Monoclonal increases in serum proteins include conditions such as Waldenström macroglobulinemia (characterized by an increase in monoclonal IgM antibodies), multiple myeloma (proliferation of malignant plasma cells, leading to the production of monoclonal immunoglobulins usually IgG or IgA), and cryoglobulinemia (this can result in monoclonal or mixed cryoglobulins, which are abnormal proteins that can precipitate and cause vascular issues, often associated with hepatitis C infection).

The conditions that involve a polyclonal increase in serum proteins include high-titer rheumatoid factor (seen in autoimmune conditions, especially rheumatoid arthritis), Sjögren’s syndrome (where the use of intravenous immunoglobulin (IVIg) infusion therapy can lead to transient polyclonal increases in serum proteins), IgG4-related disease (a rare autoimmune condition marked by the polyclonal elevation of IgG4 levels in serum and tissue infiltrates), and HIV infection (chronic viral infections such as HIV can cause polyclonal hypergammaglobulinemia as part of the immune response). The RBC-related conditions causing HVS are polycythemia vera, cyanotic heart disease, and HbSS (sickle cell anemia). The WBC-related conditions causing HVS are CLL, CML, and ANLL (acute non-lymphoblastic leukemia) [39].

In the 1950s, plasmapheresis was initially performed to address macroglobulinemia, specifically aimed at reversing retinopathy and other manifestations of hyperviscosity syndrome (HVS). Plasmapheresis remains effective in the short-term management of macroglobulinemia-related complications. It is particularly beneficial in alleviating visual symptoms associated with macroglobulinemia, preventing blindness [40].

HVS is considered an oncologic emergency due to its potentially life-threatening complications. Immediate intervention is crucial to prevent severe complications, including life-threatening thromboembolic events, myocardial infarction, and ischemic damage to vital organs. Short-term symptom control is vital in managing HVS. Plasmapheresis is the most promising short-term treatment option. It can rapidly reduce the blood’s viscosity, and it is safe to be performed daily until the symptoms are resolved. It can decrease the blood’s viscosity by 20–30%. In addition to short-term symptom control, the long-term management focuses on addressing the underlying hematologic condition causing HVS [38].

#### 4.2.5. Autoimmune Hemolytic Anemia (AIHA)

Autoimmune hemolytic anemia, often of unknown cause, involves the destruction of red blood cells by autoantibodies. The first-line treatment involves corticosteroids; however, some patients may not respond to treatment, and it can cause significant side effects in others. One case report discussed a patient with severe IgG subtype hemolytic anemia who did not respond to the standard treatment but showed improvement after five sessions of TPE. Another study comparing the treatment of severe autoimmune hemolytic anemia with TPE versus no TPE found that TPE was associated with adverse outcomes. Among 255 AIHA patients who received TPE and 4937 who did not, those who underwent TPE experienced higher rates of adverse events, including mortality, mechanical ventilation, shock, stroke, infections, kidney injury, and hemodialysis [41]. The American Society of Apheresis (ASFA) categorizes TPE for AIHA as unclear in its efficacy but can be considered in refractory cases pending a definitive treatment such as splenectomy, particularly in fulminant hemolysis [41,42].

#### 4.2.6. Systemic Lupus Erythematosus (SLE)

SLE, or systemic lupus erythematosus, is a condition marked by autoantibodies affecting multiple tissues and organ systems, causing widespread inflammation. It can result in a condition known as systemic lupus-erythematosus-associated autoimmune hemolytic anemia (SLE-AIHA). This condition can cause severe hemolysis. The standard approaches include glucocorticoids and immunosuppressive agents; however, the response is slow, necessitating alternate treatment modalities. One study suggested that TPE therapy could be a potential solution, especially in patients with life-threatening hemolysis [43]. TPE can help with removing the offending antibodies and may be beneficial in limiting the hospital stay length, improving the recovery time, and avoiding the use of steroids. Larger prospective studies are needed to support its efficacy [43].

Alveolar hemorrhage, a rare but life-threatening complication in systemic lupus erythematosus (SLE) patients, is characterized by symptoms such as anemia, hemoptysis, lung infiltrates, and respiratory failure. In 2 cases, despite using high-dose steroid therapy, the patients did not improve, so TPE was used as a last resort. Plasma exchange removes immune complexes implicated in SLE complications. Rebound autoantibody production and infection risks are some of the concerns post-TPE. Immunosuppressive drugs were combined with plasma exchange and resulted in marked improvements. The patient was eventually weaned off the ventilatory support and their renal function was restored. Conversely, previous reviews have not demonstrated a clear survival benefit with TPE. However, due to the life-threatening nature of DAH in SLE, TPE continues to be considered a treatment modality in the ASFA guidelines, despite limited evidence supporting its use [44,45].

#### 4.2.7. Cryoglobulinemia-Related Vasculitis

Cryoglobulinemia-related vasculitis is caused by immunoglobulins that precipitate at low temperatures and can cause arthralgias, purpuras, ulcers, systemic vasculitis, and peripheral neuropathies. It is often associated with chronic hepatitis C virus infection or monoclonal gammopathies. The treatment is often multipronged and can include steroids, rituximab, chemotherapy, and antiviral agents, depending on the underlying etiology; however, the response to treatment can be sluggish. Since TPE removes cryoglobulins, it can serve as an initial intervention and bridge to traditional therapies. Extensive randomized studies will be needed for further support of this treatment paradigm. For HCV-associated cases, direct-acting antivirals have shown some success in managing complications, although TPE remains critical for severe manifestations. TPE has also shown some benefit in assuaging the renal complications of Waldenström’s disease. It is, therefore, important that prospective multicenter, randomized studies are performed to support the role of TPE in treating conditions such as severe cryoglobulinemia-related vasculitis and monoclonal gammopathies, given their inherent complexities [46].

### 4.3. Other Conditions Requiring TPE in the ICU

#### 4.3.1. Drug-Induced Toxic Epidermal Necrolysis (TEN)

TEN is a rare and life-threatening skin condition characterized by skin necrosis (death) and peeling of the epidermis. It is most commonly induced by certain drugs, including anticonvulsants, allopurinol, antibiotics, and NSAIDs. Unlike Stevens–Johnson syndrome (SJS), TEN involves a larger portion of the body surface area (>30%), and it often affects mucosal surfaces such as the eyes and mouth [47].

TEN has a high mortality range, with rates ranging from 25% to 70%, which underscores the need for early intervention and effective treatments. Plasmapheresis is a medical procedure that involves removing and replacing a patient’s plasma, which contains various substances, including antibodies and inflammatory mediators. The exact pathogenesis of TEN and its relationship with plasma exchange are not entirely clear. Some studies have suggested some benefit in treating TEN while others have shown little to no benefit when compared against alternative treatments [48]. 

In one study involving seven patients, all of them had successful plasmapheresis treatments with no new skin lesions, and no adverse effects were reported. PE also provided improvements in pain and necrolysis, which indicates its potential as a treatment option for TEN [48].

In another study involving a 41-year-old female patient who was unresponsive to corticosteroids, DFPP (double filtration plasmapheresis) was administered. The patient showed a positive response, with re-epithelization occurring after the second session of DFPP [49]. 

In a case study involving a 40-year-old woman undergoing anti-tuberculosis therapy who subsequently developed toxic epidermal necrolysis (TEN), the plasmapheresis therapy proved to be successful. The commonly implicated anti-TB drugs associated with TEN are isoniazid and streptomycin. This patient, who had type 2 diabetes mellitus (DM) and tuberculosis, had been receiving anti-TB drugs for a month when she presented to the hospital. Despite discontinuing the anti-TB drugs and administering steroids, no improvement in her condition was noted. In response, plasmapheresis was initiated along with the use of 5% albumin. Remarkably, re-epithelialization of the skin was observed within just three days of commencing this treatment [47].

However, a recent nationwide retrospective study in Japan published in JAMA Dermatology in 2023 found no significant benefit of plasmapheresis therapy versus IVIG as the initial treatment of choice when systemic corticosteroid treatment was shown to be ineffective in patient with Stevens–Johnson syndrome (SJS) and toxic epidermal necrolysis (TEN) [50]. In addition, the plasmapheresis-first group had higher medical costs and an increased length of hospital stay.

In summary, the decision to use plasmapheresis in TEN should only be made with caution and only after careful and exhaustive consideration of the patient’s specific condition using the available evidence as a guide. Further research will be necessary to formulate clear guidelines for its use in the management of TEN.

#### 4.3.2. Acute Liver Failure and Acute Fatty Liver of Pregnancy (ALF and ACLF)

ALF, which can occur in previously healthy individuals or in the context of acute-on-chronic liver failure (ACLF), has high mortality rates. The common causes include acetaminophen toxicity, viral hepatitis, and autoimmune hepatitis. Wilson’s disease, a condition of excess copper, can also result in acute liver failure. While liver transplantation (LT) is the definitive treatment for ALF, TPE is being explored as a bridge to recovery or LT when immediate transplantation is not possible. TPE removes various toxins, improves the patient’s hemodynamics, and blunts inflammatory responses. In a study by Gao et al., full-dose plasma exchange was used and led to improved liver function [51]. Despite controlled trials showing some measure of efficacy in improving the systemic hemodynamics, in addition to the transplant-free survival rate, its ultimate impact on overall survival remains indeterminate. 

In AFLP, urgency is paramount and delivery is recommended to occur as quickly as possible. No prospective randomized studies seem to exist for AFLP; however, the retrospective evidence suggests that the combination therapy of TPE and CRRT can be used with promising outcomes [5].

TPE is essential in the treatment of ALF and AFLP, particularly when liver transplantation is not immediately available. Although the evidence from randomized clinical control trials shows some benefit in improving certain parameters and the transplant-free survival rate, further research will be paramount in elucidating its role in overall survival. In addition, TPE’s safety and practicability make it a worthwhile tool in the ICU for life-threatening conditions [5].

#### 4.3.3. Hypertriglyceridemic Pancreatitis

Acute pancreatitis due to hypertriglyceridemia has mortality rates of up to 30% [52]. Hypertriglyceridemia is a condition characterized by elevated levels of triglycerides in the blood. It can lead to serious complications, including acute pancreatitis and cardiovascular issues. The management of hypertriglyceridemia typically involves lipid-lowering drugs and dietary restrictions. Plasmapheresis may be considered in cases of immediate need, such as acute pancreatitis. One study recruited patients with acute pancreatitis related to hypertriglyceridemia [53]. The criteria for inclusion were triglyceride levels exceeding 1000 mg/dL and the presence of clinical, radiological, or analytical evidence of pancreatitis (elevated serum amylase and lipase levels or urine an amylase level three times the upper normal limit). The study included 11 patients, eight of whom had successful outcomes, while three unfortunately died due to severe pancreatitis upon admission [53]. Plasmapheresis was found to be successful in reducing triglyceride levels, with minimal adverse effects such as hypervolemia, which was managed with diuretics. Plasmapheresis helped remove chylomicrons, proteases, and pro-inflammatory cytokines in patients with acute pancreatitis, reducing hyperviscosity. Despite plasmapheresis showing some promise in the trial, it would be difficult to extrapolate these results given the small number of patients in this trial and lack of randomized controlled trials (RCTs) in general. Plasmapheresis as a treatment paradigm for hypertriglyceridemia-induced pancreatitis can be considered but should be approached with caution and on a case-by-case basis [53].

## 5. Discussion

### 5.1. Challenges and Limitations

Plasmapheresis can be performed in hospital settings, intensive care units, and specialized apheresis centers. Several challenges and limitations are associated with performing plasmapheresis in ICU settings, particularly considering the hemodynamic instability of ICU patients. Many ICU patients are already hemodynamically unstable due to their underlying medical conditions, and plasmapheresis can further cause hypotension and electrolyte imbalance, most notably hypocalcemia and hypokalemia, leading to cardiac arrhythmias [54,55]. The use of citrate to maintain the patency of apheresis circuits can also worsen hypocalcemia. An allergic reaction to an anaphylaxis reaction due to donor plasma and red blood cell exposure is also possible. ICU patients are also prone to coagulopathies, and patients receiving plasmapheresis are likely to have reduced hemoglobin, platelet, and fibrinogen levels, which can further worsen the coagulopathies [54,55,56]. In addition, plasmapheresis is an invasive procedure requiring a central venous catheter; there is always a risk of hospital-acquired infections [57]. Rarely, symptoms resembling anaphylaxis, such as flushing, hypotension, abdominal cramping, and other gastrointestinal symptoms, have been reported in patients who had taken an ACE inhibitor within 24 to 30 h of plasmapheresis and were receiving albumin for fluid replacement [58]. Plasmapheresis also requires specialized equipment and trained healthcare personnel, which adds to the cost of the procedure. 

The safety of plasmapheresis relies significantly on the experience of the therapeutic team and the severity of the disease being treated. Plasmapheresis is often considered in critical situations, mainly when there is a high risk of irreversible organ damage or death without immediate intervention. The timing of when to start the plasmapheresis is also crucial. Generally, patients should have received appropriate initial medical treatment with an inadequate response to or worsening symptoms despite this initial treatment before considering plasmapheresis [5]. However, it is important to note that when plasmapheresis is initiated, any concurrent treatments for the underlying disease can also be removed. Therefore, it is advisable to administer medications after the plasmapheresis procedure [1]. There are no specific guidelines and protocols to decide when exactly plasmapheresis should be initiated in ICU patients. Therefore, the decision to perform plasmapheresis should be made on a case-by-case basis [54].

Plasmapheresis should be avoided in patients with hemodynamic instability, septicemia, or a history of prior anaphylaxis reaction to one of the components of plasmapheresis, such as albumin, FFP, or heparin [54]. Continuous monitoring and fluid and electrolyte replacement are essential to prevent potential complications from plasmapheresis. Hypotension should be avoided. FFP or albumin-based fluid should be administered to prevent the filtration of the proteins being transfused. Albumin-based fluid is superior to FFP in reducing the potential side effects [59]. The citrate-related side effects can be overcome via prophylactic calcium gluconate administration during the procedure. The use of 10 mL of 10 percent calcium gluconate per liter of non-plasma fluid may prevent citrate toxicity [60]. Other non-life-threatening allergic reactions can be managed using supportive treatments. The blood flow rates should be adjusted to the patient’s tolerability. A higher blood flow rate will decrease the time needed to achieve the desired plasma exchange but may not be well tolerated [57].

The data on plasmapheresis in the ICU are limited and often conflicting, given the significant reliance on case series and case reports. In addition, new data have challenged the benefit of TPE in some cases. One example was the PEXIVAS study, which led the American Society of Apheresis to downgrade ANCA-associated vasculitis from a category 1 to a category II indication, further exemplifying the changing understanding and conflicting landscape of data regarding its effectiveness [57]. The scarcity of studies emphasizes the large gap in our understanding and presents difficulties in establishing clear guidelines and best practices; therefore, extrapolating findings from these cases to a broader population can be potentially detrimental.

### 5.2. Future Directions

This manuscript underscores the current role of therapeutic plasma exchange (TPE) in various medical conditions. Most of the evidence supporting its use in explorative indications comes from case reports, small case series, and retrospective or uncontrolled prospective data. There is an urgent need to perform well-designed trials to establish the clinical applicability of TPE in these contexts. The low incidence rates for some of the diseases mentioned earlier will likely require international collaboration to conduct robustly designed studies [1]. The timing, off-target effects, and dosing will also need to be addressed. The research should identify parameters, including imaging markers and biomarkers, to predict TPE’s effectiveness and establish reliable stopping criteria. A TPE registry could enhance our understanding of the challenges in rarer conditions and demographic-based differences [1]. Collecting blood before and after plasmapheresis and assessing differences in inflammatory markers (C-reactive protein (CRP)), coagulation profiles, and alterations in immune cells may help when analyzing TPE’s impacts on inflammation and host immune responses [1].

Therapeutic drug monitoring should be used to evaluate drug levels before and after TPE to inform the dosing protocol, particularly for essential medications such as immunosuppressants, as they may be completely removed by the process. The doses may need to be increased, repeated, or even administered post-transfusion in these cases [1].

Systematic, multicentric standardized data collection on allergic reactions, anticoagulation, and replacement fluid choice is necessary to assess risk factors. Comparing various plasma types may illuminate the immunological basis for adverse reactions. Multicenter sepsis and COVID-19 trials involving TPE are forthcoming, as are studies on neurological conditions, trauma-associated organ failure, autoimmunity, and transplantation. Molecular questions may be addressed through the use of animal disease models. The future research must also incorporate patient perspectives to assess the patient-centered outcomes and benefit–risk balance comprehensively [1].

## 6. Conclusions

Therapeutic plasma exchange is a vital therapeutic procedure in the intensive care unit (ICU) for managing multiple disorders, some of which include the hematologic, autoimmune, and neurological systems. This technique involves removing, treating, and reinfusing a patient’s plasma. Plasmapheresis plays a pivotal role in rapidly removing harmful antibodies, immune complexes, and inflammatory factors from the bloodstream in the ICU. The challenges include possible worsening of the hemodynamics, electrolyte imbalances, coagulopathies, and infection risks. It is, therefore, important to carefully assess the patient appropriateness and timing of initiation. We have highlighted the use of TPE in some disease states and its potential benefits. There is a need for more well-designed trials, international collaboration, and research into the predictive parameters to augment its applicability and safety. 

## Figures and Tables

**Table 1 medicina-59-02152-t001:** Indications for therapeutic plasma exchange (TPE) (adapted from ASFA 2023 guidelines).

System	Lines	Diagnosis	Specific Condition
Neurological disorders	First-line	Acute inflammatory demyelinating polyneuropathy	
		Chronic acquired demyelinating polyneuropathies, IgG/IgA/IgM-related	
		Chronic inflammatory demyelinating polyradiculoneuropathy	
		Myasthenia gravis	Acute, short-term treatment
		N-methyl-D-aspartate receptor antibody encephalitis	
	Second-line	Lambert–Eaton myasthenic syndrome	
		Multiple sclerosis	Acute attack/relapse; long-term treatment
		Neuromyelitis optical spectrum disorder	Acute attack/relapse
		Pediatric autoimmune neuropsychiatric disorders	PANDAS/PANS, exacerbation
		Steroid-responsive encephalopathy associated with autoimmune thyroiditis	
Hematological disorders	First-line	Catastrophic antiphospholipid syndrome	
		Hyperviscosity in hypergammaglobulinemia	Prophylaxis for rituximab; symptomatic hyperviscosity syndrome
		Thrombotic microangiopathy, complement-mediated	Factor H autoantibody-related only
		Thrombotic microangiopathy, drug-induced	Ticlopidine-related only
		Thrombotic microangiopathy, thrombotic thrombocytopenic purpura	
	Second-line	Lambert–Eaton myasthenic syndrome	
		Multiple sclerosis	Acute attack/relapse; long-term treatment
		Neuromyelitis optical spectrum disorder	Acute attack/relapse
		Pediatric autoimmune neuropsychiatric disorders	PANDAS/PANS, exacerbation
		Steroid-responsive encephalopathy associated with autoimmune thyroiditis	
Transplantation-associated complications	First-line	Transplantation, kidney, ABO-compatible	Antibody-mediated rejection; Desensitization/prophylaxis, living donor
		Transplantation, kidney, ABO-incompatible	Desensitization, living donor
		Transplantation, liver	Desensitization, ABOi, living donor
	Second-line	Transplantation, heart	Desensitization; rejection prophylaxis
		Transplantation, hematopoietic stem cell, ABO-incompatible	Major ABO incompatible: HPC(M); HPC(A)
		Transplantation, kidney, ABO-incompatible	Antibody-mediated rejection
Renal disorders	First-line	Antiglomerular basement membrane disease	Diffuse alveolar hemorrhage; dialysis-independent disease
		Focal segmental glomerulosclerosis	Recurrent in kidney transplant
Hepatic disorders	First-line	Acute liver failure (TPE-HV preferred over regular TPE)	Other than acute fatty liver of pregnancy
		Wilson disease, fulminant	
Other Systems	Second-line	Systemic lupus erythematosus	
		Thyroid storm	
		Familial hypercholesterolemia	
		Phytanic acid storage disease	
		Hepatitis B related polyarteritis nodosa vasculitis	
		Voltage-gated potassium channel antibody-related diseases	
		Mushroom poisoning	

PANDAS, pediatric autoimmune neuropsychiatric disorders associated with streptococcal infections; PANS, pediatric acute-onset neuropsychiatric syndrome; HPC(M), hematopoietic progenitors from bone marrow; HPC(A), allogeneic hematopoietic progenitor cell; TPE-HV, therapeutic plasma exchange—high volume.

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
