# Peer review of "Plasmapheresis in the ICU"

_medicina, 2023, doi:10.3390/medicina59122152_

Round 1
Reviewer 1 Report
Comments and Suggestions for Authors
Dear Editors,
The topic dealt with by the authors of the paper is interesting not only to the scientific community in the field of medicine, but also to the wider community, because it points to the needs of modern medicine to find new methods not only in treatment, but also in prevention. The reason for this lies in the fact that an increasing number of acute diseases are characterized by endotoxicosis, which is the cause of multi-organ insufficiency, which are medically serious conditions. Doctors are increasingly faced with this kind of problem in their medical practice, and the most recent example is the COVID-19 pandemic. In such cases, modern medicine has neither surgery, nor the most effective drugs, nor effective therapeutic treatment protocols that lead to healing.
For many years, efferent therapy methods have been applied to these diseases. Unfortunately, the results of the mentioned method are known only through reports on individual cases, and on the other hand, there is no consensus among all experts about the benefits of efferent therapy, as the authors point out through the cited references in the paper.
CONCLUSION
The proposed publication has its scientific significance, which is reflected through the analysis of a large number of references that indicate the following:
- the research topic indicates the indications and therapeutic potential of plasmapheresis as a form of efferent therapy in various acute and chronic conditions,
- gives basic guidelines to the scientific community about the needs of well-designed clinical research, the results of which would be useful for creating protocols for the therapeutic and/or preventive application of efferent therapy in medicine,
- the need for education about the potential of efferent therapy as a "potential powerful weapon" in the fight against biological warfare.
Author Response
Thank you for your detailed response and assessment.
Reviewer 2 Report
Comments and Suggestions for Authors
Line 34: Based on historic data while efficacy was comparable , centrifugal process was reported to be more time -efficient.
Line 67: Table 1 not provided for review.
Line:87: Please change to”also been investigated”
Line 136: Please provide in supplementary material : What is Neuropathy impairment score.
Line 139: by removing autoantibodies sounds more exact that harmful components. Also this is repetitive as already mentioned in previous paragraph (line 127-128)
Line 166-168: Repetition of 157-159. Please combine the reference cited(Dau PC et.al) to avoid repetitive sentences.
Line 212: 217: Please explain this better. Does the baseline impairment and PLEX delay positively or negatively associated with complete improvement. Also please explain what was considered good and poor secondary outcome in the study.
Line 221: thrombi or microthrombi formation is more appropriate than “clot formation”
Please add references to Line 272(MEPEX trial) and Line 273(PEXIVAS trial)
TMA: (line 280-289)
It is a spectrum of disorder where there is hemolytic anemia, organ damage due to underlying thrombotic events. CAPS, TTP, HUS is all considered under this umbrella along with drug induced TMAs. Please clarify this paragraph as to which specific TMA you are addressing here or use it as introduction for all hematologic disorders under this umbrella needing TPE.
Line 292: add reference to the triad described.
Section 4.3: TEN:
Ref:Evaluation of plasmapheresis vs Immunoglobulin as First Treatment After Ineffective Systemic Corticosteroid Therapy for Patients With Stevens-Johnson Syndrome and Toxic Epidermal Necrolysis. JAMA dermatol 2023
Conclusion: Plasmapheresis did not show any added benefit but increased cost, ,longer hospital stay.
Hence suggesting use of TPE in TEN should be considered with caution as not validated.
Hypertriglyceridmia with pancreatitis:
In addition to lack of RCT;the trial itself had only 11 patients so extrapolation of results should be done with abundant caution
Line 492-493: Some of the indications for plasmapheresis is in conditions with thrombotic predisposition. So when mentioning coagulopathy as a contraindication please be very clear on the bleeding risk profile and when to consider that. (eg CAPS can have bleeding and clotting risk)
Future Directions:
Line 506-507: Underscores potential of TPE in future- I would consider replacing with current role of TPE (potential is typically used for well validated and possibility of increasing scope).
Line 516-520: Please explain what you propose with bio banking (of what specimen pre/post both plasma or other tissue) and assessment of host immunity. Also many of the drugs can be removed with TPE so how do you envision drug dosing to impact this effect?
Many of the sentences are overall repetitive in the manuscript- I have highlighted a couple. Please try to make the narration crisper. Also acknowledge limitations where data is lacking - When only few case series/case reports are available as evidence in literature or data is controversial it is essential to acknowledge this as extrapolation to a broader population can be detrimental. This is not addressed in discussion or under specific disease conditions.
Comments on the Quality of English LanguageAdequate
Author Response
Line 34: Based on historic data while efficacy was comparable , centrifugal process was reported to be more time -efficient.
Response : Adjusted (line 37-38)
Line 67: Table 1 not provided for review..
Response : Table added on page 3 of 16
Line:87: Please change to”also been investigated”
Response : adjusted, line 90
Line 136: Please provide in supplementary material : What is Neuropathy impairment score.
Reponse : added to supplementary materials
Line 139: by removing autoantibodies sounds more exact that harmful components. Also this is repetitive as already mentioned in previous paragraph (line 127-128)
Response : repetitive sentence removed. see line 146.
Line 166-168: Repetition of 157-159. Please combine the reference cited(Dau PC et.al) to avoid repetitive sentences.
Response : sentence adjusted and moved to 163-165
Line 212: 217: Please explain this better. Does the baseline impairment and PLEX delay positively or negatively associated with complete improvement. Also please explain what was considered good and poor secondary outcome in the study.
Response: paragraph adjusted including aforementioned clarifications. lines 211-224
Line 221: thrombi or microthrombi formation is more appropriate than “clot formation”
Response: changed to microthrombi. line 228
Please add references to Line 272(MEPEX trial) and Line 273(PEXIVAS trial)
Response : references added. lines 280 and 283 respectively
TMA: (line 280-289)
It is a spectrum of disorder where there is hemolytic anemia, organ damage due to underlying thrombotic events. CAPS, TTP, HUS is all considered under this umbrella along with drug induced TMAs. Please clarify this paragraph as to which specific TMA you are addressing here or use it as introduction for all hematologic disorders under this umbrella needing TPE.
Response: I agree with your assessment. will remove paragraph altogether to avoid redundancy
Line 292: add reference to the triad described.
response : see line 290. reference added
Section 4.3: TEN:
Ref:Evaluation of plasmapheresis vs Immunoglobulin as First Treatment After Ineffective Systemic Corticosteroid Therapy for Patients With Stevens-Johnson Syndrome and Toxic Epidermal Necrolysis. JAMA dermatol 2023
Conclusion: Plasmapheresis did not show any added benefit but increased cost, ,longer hospital stay.
Hence suggesting use of TPE in TEN should be considered with caution as not validated.
Response: Thank you. adjusted section and incorporated the 2023 study. Included summary sentences 422-425 highlighting need for caution and further research
Hypertriglyceridmia with pancreatitis:
In addition to lack of RCT;the trial itself had only 11 patients so extrapolation of results should be done with abundant caution
response: adjusted paragraph and insinuated necessity for a cautionary approach in lines 464-468
Line 492-493: Some of the indications for plasmapheresis is in conditions with thrombotic predisposition. So when mentioning coagulopathy as a contraindication please be very clear on the bleeding risk profile and when to consider that. (eg CAPS can have bleeding and clotting risk)
response : removed coagulopathy so as to avoid confusion
Future Directions:
Line 506-507: Underscores potential of TPE in future- I would consider replacing with current role of TPE (potential is typically used for well validated and possibility of increasing scope).
response : I agree. Adjusted (line 516)
Line 516-520: Please explain what you propose with bio banking (of what specimen pre/post both plasma or other tissue) and assessment of host immunity. Also many of the drugs can be removed with TPE so how do you envision drug dosing to impact this effect?
Adjusted paragraph 535-541.
Many of the sentences are overall repetitive in the manuscript- I have highlighted a couple. Please try to make the narration crisper. Also acknowledge limitations where data is lacking - When only few case series/case reports are available as evidence in literature or data is controversial it is essential to acknowledge this as extrapolation to a broader population can be detrimental. This is not addressed in discussion or under specific disease conditions.
response: Attempted to summarize and remove some redundant information. I also added 514-522 to the discussion to help reiterate those concerns about limited and conflicting data.
Round 2
Reviewer 2 Report
Comments and Suggestions for Authors
Thank you for revising the manuscript. My few comments as below:
Line 84: Myasthenia Gravis repeated twice.
Table:1 alignment to be formatted.
If possible please provide EDSS score also in supplementary material.
Line:399( instead of reference number says new reference)
Line 421-422: Says again new reference without reference number.
Line :456: Please provide reference for the study cited for hypertriglyceridemic pancreatitis.
Line 460: Please provide reference (Is it similar to the one cited in Line 456?)
Author Response
Line 84: Myasthenia Gravis repeated twice.
corrected.
Table:1 alignment to be formatted
will make further adjustments to alignment during publication process
If possible please provide EDSS score also in supplementary material.
added to supplementary materials
Line:399( instead of reference number says new reference)
adjusted ( line 403)
Line 421-422: Says again new reference without reference number.
adjusted ( lines 424 to 426)
Line :456: Please provide reference for the study cited for hypertriglyceridemic pancreatitis.
adjusted (461)
Line 460: Please provide reference (Is it similar to the one cited in Line 456?)
adjusted (465)
Also adjusted GBS, MS and NOSD sections of neurological disorders
Thank you
